# Multiscale Convolutional Neural Network Based on Channel Space Attention for Gearbox Compound Fault Diagnosis

**DOI:** 10.3390/s23083827

**Published:** 2023-04-08

**Authors:** Qinghong Xu, Hong Jiang, Xiangfeng Zhang, Jun Li, Lan Chen

**Affiliations:** College of Intelligent Manufacturing and Industrial Modernization, Xinjiang University, Urumqi 830017, China; xqh@stu.xju.edu.cn (Q.X.); xjuzxf@xju.edu.cn (X.Z.); junli@stu.xju.edu.cn (J.L.); 18324012616@163.com (L.C.)

**Keywords:** gearboxes, compound faults, multiscale feature extraction, attentional mechanisms, multilabel classification

## Abstract

Gearboxes are one of the most widely used speed and power transfer elements in rotating machinery. Highly accurate compound fault diagnosis of gearboxes is of great significance for the safe and reliable operation of rotating machinery systems. However, traditional compound fault diagnosis methods treat compound faults as an independent fault mode in the diagnosis process and cannot decouple them into multiple single faults. To address this problem, this paper proposes a gearbox compound fault diagnosis method. First, a multiscale convolutional neural network (MSCNN) is used as a feature learning model, which can effectively mine the compound fault information from vibration signals. Then, an improved hybrid attention module, named the channel–space attention module (CSAM), is proposed. It is embedded into the MSCNN to assign weights to multiscale features for enhancing the feature differentiation processing ability of the MSCNN. The new neural network is named CSAM-MSCNN. Finally, a multilabel classifier is used to output single or multiple labels for recognizing single or compound faults. The effectiveness of the method was verified with two gearbox datasets. The results show that the method possesses higher accuracy and stability than other models for gearbox compound fault diagnosis.

## 1. Introduction

Gearboxes are one of the most widely used speed and power transfer elements in rotating machinery and play a vital role in the manufacturing industry. Gearbox failures can lead to unplanned shutdowns, causing substantial economic losses and even significant casualties. Therefore, in order to prevent accidents and ensure the efficient operation of mechanical systems, it is essential to carry out fault diagnosis of gearboxes [1,2].

Among the gearbox fault diagnosis methods, intelligent fault diagnosis is one of the most widely used methods. Hajnayeb et al. [3] designed a fully connected neural network to classify four operating conditions using gearbox vibration signals. D.J. Bordoloi et al. [4] used different optimization algorithms to optimize support vector machine parameters for the multi fault classification of gears using time-frequency vibration data. Hu et al. [5] proposed a fault diagnosis method based on multiscale dimensionless metrics and random forest that can identify different types of faults with an average accuracy of 95.58%. Traditional intelligent fault methods perform feature extraction using signal processing [6,7,8] and then use shallow machine learning [9,10,11] models for fault identification. However, these methods require extensive diagnostic expertise and signal-processing calculations during feature extraction, which are time-consuming and dependent on expert experience [12]. Moreover, as its feature extraction and classification processes are designed independently, unsynchronized model optimization will consume significant time and limit its diagnostic performance [13].

Driven by artificial intelligence technology, deep learning is also widely used in fault diagnosis. Compared with traditional intelligent fault methods, deep learning-based methods are free from reliance on expert knowledge and signal pre-processing methods and have obvious advantages in the face of massive data. Zhang et al. [14] proposed a wide kernel convolutional neural network that uses wide convolutional kernels in the first convolutional layer to suppress high-frequency noise and uses small convolutional kernels to extract deep features. The results showed that the model is robust to variable load conditions and noise. Xia et al. [15] proposed a multi-sensor-based convolutional neural network diagnosis method to fuse multiple sensor signals for gearbox compound fault diagnosis. These methods have achieved fruitful results in fault diagnosis. However, fault-related information contained in vibration signals is usually distributed on multiple scales, and this must be considered. Most only perform single-scale feature extraction and cannot obtain multiscale fault information.

In recent years, some scholars have attempted to construct multiscale convolutional neural network models for capturing the multiscale features of vibration signals. Chen et al. [16] combined a multiscale convolutional neural network with a long- and short-term memory network for fault diagnosis of rolling bearings. Jiang et al. [17] constructed a multiscale convolutional neural network to identify different health states of wind turbine gearboxes. K.N. Ravikumar et al. [18] proposed a multilayer long and short-term memory multiscale deep residual learning fault diagnosis model in internal combustion engine gearboxes. Although these multiscale convolutional neural networks are enriched with fault feature information, they combine the captured multiscale features; precisely, it ignores the different importance of the scale features learned in different branches for the final diagnosis decision, and even some of this misinformation can mislead the diagnosis decision [19].

Attention mechanisms consider weighting effects, i.e., mapping relationships between inputs and outputs, which can enhance critical features and weaken redundant components [20,21]. Therefore, introducing the attention mechanism into the diagnostic model can improve the validity and reliability of the method. Jang et al. [22] introduced spatial attention into autoencoders and designed attentional self-encoders to learn and calibrate the location information in the potential space. Plakias et al. [23] introduced spatial attention into a dense convolutional neural network to improve the feature extraction capability of the model and achieve recognition of bearings with different loss levels. Li et al. [24] designed a fusion strategy based on a channel attention mechanism to obtain more fault-related information during the fusion of multisensor data features. Xie et al. [25] constructed an improved convolutional neural network incorporating a channel attention mechanism for fault diagnosis of diesel engine systems. Huang et al. [26] designed a hybrid attention method to adaptively select important features through tandem spatial and channel attention. They proposed a shallow multiscale CNN with mixed attention for bearing fault diagnosis. Ye et al. [27] presented the same spatial attention as the SE-Net process but with different dimensions. They connected the channel attention in series with the proposed spatial attention for bearing fault diagnosis. Despite the success of traditional intelligent diagnosis methods, there are still some limitations, mainly in the form of:

(1) For spatial attention, the information interaction between channels is ignored because the features of each channel are treated equally. Similarly, channel attention is directly global to the information within the channel and tends to ignore the information interactions in space [28].

(2) In the face of compound faults, conventional classifiers can only output one label instead of multiple labels, meaning it is impossible to classify compound faults into two or more single faults. Compound faults are considered a special independent failure mode in fault diagnosis [29]. For single-class faults and normal states, there exist $2^{n}-\left(n+1\right)$ classes of compound fault forms [30]. The increase in fault types causes the dimensionality of the deep learning model’s output layer to increase, leading to more network parameters and training difficulties. To address the above problems, this paper proposes a channel–space attention multiscale convolutional neural network (CSAM-MSCNN) based on the composite fault diagnosis method for gearboxes.

The main contributions of this paper are as follows:

(1) A channel–space attention module (CSAM) is proposed to solve the side effects of the bottleneck structure in hybrid attention. Then a CSAM-MSCNN is designed based on the CSAM and multiscale convolutional neural network (MSCNN), which can effectively solve the feature redundancy problem brought by multiscale feature fusion. In addition, the CSAM-MSCNN also takes into account the interaction of channel information and the interaction of spatial information.

(2) For solving the deficiency of a traditional fault diagnosis mode in a compound fault, a gearbox compound fault diagnosis method is proposed by combining a multilabel classifier and CSAM-MSCNN, which can decouple the compound fault into multiple single faults.

(3) Two gearbox datasets are leveraged to validate the compound fault diagnosis capability of the proposed method. The experimental results show that the proposed method has higher prediction accuracy for compound fault than the classical diagnosis methods.

The rest of the paper is organized as follows: Section 2 details the basic theory of convolutional neural networks and multilabel classification. Section 3 presents the proposed CSAM-MSCNN and the intelligent fault diagnosis method. In Section 4, the feasibility and superiority of the method are verified by two experimental studies, and the results are compared with other intelligent diagnostic methods using the same dataset. Conclusions and future research work are discussed in Section 5.

## 2. Theoretical Backgrounds

### 2.1. Convolutional Neural Network

A CNN, a supervised deep learning method, has achieved excellence in speech recognition, image classification, and target detection. Typically, CNNs contain a convolutional layer, an activation layer, a pooling layer, a classification layer, and some techniques to improve the model’s generalization ability, such as batch normalization.

### 2.2. Convolutional Layer

The convolutional layer is the core component of the CNN model, enabling local connectivity and weight sharing through convolutional kernels. The convolution kernel slides along the input feature map and performs convolution operations with the data in the perceptual field to achieve feature extraction. The specific arithmetic formula is as follows:(1)$$y_{j}^{l+1}=\sum_{i}^{k}x_{i}^{l}w_{ij}^{l}+b_{j}^{l}$$
where $x_{i}^{l}$ is the input at the first level, $w_{ij}^{l}$ is the weight matrix, $b_{j}^{l}$ is the bias, and $y_{j}^{l+1}$ is the output in the first level.

### 2.3. Batch Normalization Layer

Batch normalization (BN) is used to reduce the internal covariance bias and accelerate the training of deep neural networks. The BN layer is usually added after the convolutional or fully connected layer and before the activation function. The mathematical formula for the BN layer is as follows:(2)$$\mu =\frac{1}{N_{batch}}\sum_{s=1}^{N_{batch}}x_{s}$$
(3)$$\sigma^{2}=\frac{1}{N_{batch}}{\sum_{s=1}^{N_{batch}}\left(x_{s}-\mu \right)}^{2}$$
(4)$${\hat{x}}_{s}=\frac{x_{s}-\mu }{\sqrt{\sigma^{2}+\varepsilon }}$$
(5)$$y_{s}=\gamma {\hat{x}}_{s}+\beta$$
where $N_{batch}$ denotes the number of batch data, $x_{s}$ denotes the $s_{th}$ input, μ and $\sigma^{2}$ denote the mean and variance of the batch data, respectively, ε denotes a constant close to 0 but greater than 0, ${\hat{x}}_{s}$ denotes the result of data normalization, γ and β denote the parameters that the network can learn, and $y_{s}$ denotes the $s_{th}$ output of the data after BN.

### 2.4. Activation Function

The activation function enables the network to gain nonlinear representation capability, enhances the feature representation of the model, and maps otherwise linearly indistinguishable multidimensional features to another space, making the learned features easier to distinguish. In recent years, rectified linear units (ReLU) have been widely used as activation functions to accelerate the convergence of neural networks. The ReLU is calculated as follows:(6)$$f\left(x\right)=max\left(0,x\right)$$

### 2.5. Pooling Layer

The pooling layer is usually placed after the convolutional layer to reduce the number of feature dimensions and parameters and to prevent network overfitting. A typical pooling layer includes maximum pooling and average pooling with the following operations:(7)$$p_{i}^{l+1}\left(j\right)=\underset{\left(j-1\right)W+1\leq t\leq jW}{max}\left\{a_{t}^{l}\left(t\right)\right\}$$
(8)$$p_{i}^{l+1}\left(j\right)=\underset{\left(j-1\right)W+1\leq t\leq jW}{avg}\left\{a_{t}^{l}\left(t\right)\right\}$$
where $P_{i}^{l+1}$ denotes the value of the neuron after the output of the pooling layer, $a_{t}^{l}\left(t\right)$ denotes the value of the $t_{th}$ neuron in the $l_{th}$ layer, $t\in \left[\left(j-1\right)W+1,jW\right]$, W indicates the width of the pooling area.

### 2.6. Fully Connected Layer

After multiple convolutional and pooling layers, a few fully connected layers are usually secured and used to integrate the extracted features. The activation function in the fully connected layer usually uses the ReLU function. For a one-dimensional input $x_{i}^{L-1}$ of length M, the fully connected layer has N neurons and the output of each neuron can be expressed as:(9)zjL=ReLU∑i=1MxiL−1wj,iL+bjL,j=1,2,⋅⋅⋅,N
where $w_{j,i}^{L}$ is the connection weight from the $i_{th}$ neuron in the $L-1_{th}$ layer to the $j_{th}$ neuron in the $L_{th}$ layer, $x_{i}^{L-1}$ denotes the input of the $j_{th}$ neuron of the $L_{th}$ layer, $b_{j}^{L}$ denotes the bias of the $j_{th}$ neuron of the $L_{th}$ layer, M and N denote the number of neurons in layer L−1 and layer L, respectively.

### 2.7. Multilabel Classifier

The SoftMax classifier (SC) is trained using a cross-entropy loss function and is widely used in multiclassification tasks. Given a training set ${\left\{x_{i},y_{i}\right\}}_{i=1}^{M}$, where $x_{i}$ and $y_{i}$ are the $i_{th}$ sample and label, respectively, and M is the total number of samples. The SoftMax function has the following operator equation:(10)$$s_{i}=softmax\left(y_{i}^{FC}\right)=\frac{e^{y_{i}^{FC}}}{\sum_{c=1}^{C}e^{y_{c}^{FC}}}$$
where $y_{i}^{FC}$ is the output of the $i_{th}$ neuron in the last fully connected layer, $s_{i}\in s=\left[s_{1},s_{2},\cdot \cdot \cdot ,s_{C}\right]$, C is the number of categories. The label of SC can be calculated by the following equation:(11)$$label_{soft\max}=\arg\max\left(s\right)$$
where $\arg\max\left(s\right)$ denotes the index value of the maximum value in the vector s.

The cross-entropy loss function has a more vital global optimization ability and faster convergence than other loss functions. It is widely used in deep convolutional neural networks with the following equation:(12)$$J=-\frac{1}{M}\left[\sum_{m=1}^{M}\sum_{c=1}^{C}1\left\{y_{m}=c\right\}\log\left(s_{c}^{m}\right)\right]$$
where $1\left\{\cdot \right\}$ is the indicator function that returns 1 if $1\left\{\cdot \right\}$ turns out to be true and 0 if it is false.

Since the traditional SC cannot handle the multilabel problem, some changes are made in the output layer activation function, loss function, and labels. First, the Sigmoid function is chosen as the activation function for the last fully connected layer. Similarly, given the training set ${\left\{x_{i},y_{i}\right\}}_{i=1}^{M}$, where $x_{i}$ and $y_{i}$ are the $i_{th}$ sample and label, respectively, and M is the total number of samples, the Sigmoid function is calculated as follows:(13)$$\sigma_{i}=\sigma \left(y_{i}^{FC}\right)=\frac{1}{1+e^{-y_{c}^{FC}}}$$
where $y_{i}^{FC}$ is the output of the $i_{th}$ neuron in the previous fully connected layer and $\sigma_{i}$ denotes the confidence probability of the $i_{th}$ category, $\sigma_{i}\in \sigma =\left[\sigma_{1},\sigma_{2},\cdot \cdot \cdot ,\sigma_{C}\right]$, C is the number of categories. The Sigmoid function ensures that the value of the last fully connected layer is converted to between 0 and 1, and that the confidence probabilities of each category do not interact with each other.

Then, in multiclassification, a sample has only one label, while a sample in multilabel classification may contain more than one label. The multilabel classification problem is usually converted into an N binary classification problem, so the binary cross-entropy function is used as the loss function to train the CSAM-MSCNN network, which is calculated as follows:(14)$$J=-\frac{1}{C}\sum_{i=1}^{C}y_{i}\log\left(\sigma_{i}\right)+\left(1-y_{i}\right)\log\left(1-\sigma_{i}\right)$$

Finally, a class-unique thermal coding is used for the labels, i.e., the number of all single-fault types is used as the length of the label vector, and the compound fault is considered as a combination of multiple single faults so that its label exists as one at various indices.

The SC takes out the fault type with the highest confidence probability as its prediction label by the argmax function. The multilabel classifier (MLC) filters the confidence probabilities by a threshold ϕ to output predicted labels. If the confidence probability of the fault type is greater than the threshold value, the value at the index of its predicted label vector is set to 1. Otherwise, it is set to 0. The threshold value in this paper is set to 0.7.

The differences between SC and MLC are shown in Figure 1. Squares, circles, and triangles indicate three different single faults, and the overlapping forms of triangles and circles indicate compound faults. The MLC can accurately identify and decouple compound faults by outputting labels for two or more single fault types. Still, the SC classifies compound faults as error categories because it cannot decouple them.

## 3. The Proposed Method

### 3.1. Channel–Space Attention Module

The convolutional block attention module (CBAM) [21] is a typical hybrid attention method used for lifetime prediction and machinery fault diagnosis. The CBAM structure is shown in Figure 2, and it consisted of a channel attention module and a spatial attention module connected in series. CBAM combines the advantages of channel and spatial attention and considers the information interaction in the channel and space. It is better than a single channel or spatial attention module. The channel attention module in the CBAM is shown in Figure 3. First, the spatial information in the average pooling and maximum pooling aggregated feature maps was used. Then, two sets of channel attention weights were obtained through a shared fully connected layer, and the two sets of weights were combined. Finally, the final channel attention weights were received after the Sigmoid function. Although channel attention implements interactions between channels using a shared fully connected layer, the dimensionality reduction due to its bottleneck structure has side effects on channel attention prediction. It is not efficient or necessary to capture the interactions between all channels [31].

Inspired by the spatial attention module structure in the CBAM, an improved channel attention mechanism was proposed using a one-dimensional convolution instead of the original shared fully connected layer. The side effects of bottleneck structures were avoided while ensuring cross-channel information interaction and reducing model complexity. The design of the specific improved channel attention module is shown in Figure 4, and the use space attention module in this paper is shown in Figure 5. The improved Channel Attention Module (CAM) was connected in series with the original Spatial Attention Module (SAM) to form the Channel–Spatial Attention Module (CSAM) used in this paper.

The channel attention module in the CSAM was processed as follows: First, the input feature map $F_{in}$ is fed into the channel attention module. Then the average pooling and maximum pooling are used to aggregate the spatial information of $F_{in}$, and two different vectors $F_{avg}^{c}$, $F_{\max}^{c}$ describing the spatial information are obtained. The $F_{avg}^{c}$,$F_{\max}^{c}$ deformations are stitched together to obtain the spatial information synthesis description vector. After the one-dimensional convolution process, the weights are then compressed to 0~1 by the Sigmoid activation function. The channel attention weights $M_{c}$ are obtained after deformation reduction. The channel attention weight $M_{c}$ is multiplied with the input feature map $F_{in}$ to obtain the channel attention corrected feature map $F_{c}$.
(15)$$F_{avg}^{c}=Avgpool\left(F_{in}\right)$$
(16)$$F_{\max}^{c}=Maxpool\left(F_{in}\right)$$
(17)$$F_{sum}^{c}=Concatenate\left(reshape\left(F_{avg}^{c}\right),reshape\left(F_{\max}^{c}\right)\right)$$
(18)$$M_{c}=Reshape\left(\sigma \left(conv\left(F_{sum}^{c}\right)\right)\right)$$
(19)$$F_{c}=M_{c}⊗F_{in}$$

The spatial attention module in the CSAM handles the process as follows: First, the feature map $F_{c}$ corrected by channel attention is input into the spatial attention module. Then two different channel information description vectors $F_{avg}^{s}$, $F_{\max}^{s}$ are obtained using the average pooling and maximum pooling aggregation $F_{c}$ of channel information, respectively. The spatial information synthesis description vector $F_{sum}^{s}$ is obtained by stitching $F_{avg}^{s}$, $F_{\max}^{s}$ together. The spatial attention weight $M_{s}$ is obtained after a one-dimensional convolution process and then by a Sigmoid activation function. The spatial attention $M_{s}$ is multiplied by $F_{c}$ to obtain the spatial attention corrected feature map $F_{s}$, which is the output feature $F_{out}$ after correction by the CSAM module.
(20)$$F_{avg}^{s}=Avgpool\left(F_{c}\right)$$
(21)$$F_{\max}^{s}=Maxpool\left(F_{c}\right)$$
(22)$$F_{sum}^{s}=Concatenate\left(F_{avg}^{s},F_{\max}^{s}\right)$$
(23)$$M_{s}=\sigma \left(conv\left(F_{sum}^{s}\right)\right)$$
(24)$$F_{out}=F_{s}=M_{s}⊗F_{c}$$

### 3.2. Channel–Spatial Attention Multiscale Convolutional Neural Networks

The network architecture constructed in this paper is shown in Figure 6, which mainly included raw signal input, multiscale feature extraction based on channel–space attention, and fault diagnosis. First, the original data samples of the same length $X_{1},X_{2},X_{3},\cdot \cdot \cdot ,X_{n}$ were sampled from the vibration signal and fed into the network. The fault information was extracted through the feature extraction layer, and finally, all the fault information was aggregated in the classification layer, and diagnostic decisions were made.

The feature extraction layer performed feature extraction on the input sample X using three different scales. The first scale obtained the scale information of the vibration signal by averaging pooling and subsequently mined its deeper fault information using four convolution blocks. Each convolutional block contained a one-dimensional convolutional layer, a batch normalization layer, an activation function ReLU, and a highest pooling layer. The second scale directly used four convolutional blocks for feature extraction of the original scale signal. The difference is that the first layer convolution kernel used a wide convolution kernel with a sizeable sensory field to obtain global information and shallow features. Then three convolution blocks were used to mine deep features. Unlike the first scale, the third scale used maximum pooling to obtain the scale information of the vibration signal, followed by the same four convolutional blocks to learn the fault information in the scale signal. The extracted features at three scales are $f_{1}$,$f_{2}$, $f_{3}$, and fused to obtain the fused feature F.
(25)$$f_{1}=conv\left(conv\left(conv\left(conv\left(avgpool\left(X\right)\right)\right)\right)\right)$$
(26)$$f_{2}=conv\left(conv\left(conv\left(conv\left(X\right)\right)\right)\right)$$
(27)$$f_{3}=conv\left(conv\left(conv\left(conv\left(maxpool\left(X\right)\right)\right)\right)\right)$$
(28)$$F=Concatenate\left(f_{1},f_{2},f_{3}\right)$$

To solve the feature redundancy problem caused by feature fusion, the CSAM assigned different weights to the fused features. Attention-corrected features were obtained by enhancing critical fault information and suppressing useless information. Finally, the feature map was downscaled by global average pooling to get the feature map $F_{a}$. The feature map obtained from the feature extraction layer was flattened and sent to the fully connected layer. After two fully connected layers, the confidence probability P for each fault type was calculated using the Sigmoid function.
(29)$$F_{a}=GAP\left(Attention\left(F\right)\right)$$
(30)$$P=\sigma \left(Linear\left(Linear\left(Flatten\left(F_{a}\right)\right)\right)\right)$$

The main advantages of the CSAM-MSCNN proposed in this paper are as follows: (1) The feature extraction layer of the network can significantly enrich the fault feature information obtained by feature extraction of the input fault samples from three different scales. (2) Using the hybrid channel–space attention mechanism, the large amount of fault feature information extracted from the multiscale structure is weight mapped to enhance key features and weaken redundant CSAM, effectively solving the feature redundancy problem caused by multiscale feature fusion and improving the efficiency of fault diagnosis. (3) CSAM optimizes the bottleneck structure of the channel attention module in CBAM. While ensuring cross-channel information interaction, the side effects of the bottleneck structure are avoided, and the model complexity is also reduced. (4) The CSAM-MSCNN uses a Sigmoid function with a binary cross-entropy loss, which provides independent probabilities for each class and an unrestricted sum of possibilities for all classes, solving the problem that traditional intelligent fault diagnosis cannot classify compound faults into two or more single faults.

### 3.3. Fault Diagnosis Method Based on CSAM-MSCNN

The CSAM-MSCNN proposed in this paper consisted of three gearbox compound fault diagnosis parts. The first part was to collect the gearbox vibration signal using the digital acquisition system, sample it and divide the dataset. The second part was to train the CSAM-MSCNN model. The third part was the performance test evaluation of the completed training model to realize the compound fault diagnosis of the gearbox. In this method, there was no need to extract and select features manually. The original vibration signal could be used directly to achieve fault diagnosis in an end-to-end manner. Figure 7 shows the diagnostic flow chart of the CSAM-MSCNN, and the detailed steps of fault diagnosis are as follows:

(1) Acquisition of gearbox vibration signals using acceleration sensors.

(2) The sample dataset is constructed by the overlapping sampling of the original vibration signals, and the training set, validation set, and test set are divided randomly in proportion.

(3) Construct the CSAM-MSCNN model, set the network hyperparameters such as learning rate, sample batch size, and the number of training iterations, and initialize the network weights.

(4) The CSAM-MSCNN is trained using the training set samples, and the error between the network output and the sample labels is calculated using the binary cross-entropy loss function. The error is back-propagated according to the Adam gradient descent algorithm, and the network weight parameters are updated up to the set number of training iterations. The network is evaluated using the validation set for each training round, but backpropagation of errors and network weight updates are not performed.

(5) Performance test evaluation of the completed trained CSAM-MSCNN using the test set. The network prediction labels of the test set samples are compared with the sample labels to obtain the test set sample accuracy.

## 4. Experimental Verification

### 4.1. Experiment 1: Parallel Bearing Gearbox Fault Diagnosis

#### 4.1.1. Description of Experimental Data

To verify the effectiveness and superiority of the method in this paper, we used SQI’s wind turbine drive system fault diagnosis test bench to conduct compound fault diagnosis experiments and collect relevant fault data. The structure of the test bench is shown in Figure 8, which mainly contained a motor, a motor controller, a two-stage parallel shaft gearbox, a planetary gearbox, and a magnetic powder brake. A sketch of the structure of the secondary parallel shaft gearbox is shown in Figure 9. The number of gear teeth on the input and output shafts was 36 and 100, respectively, and the number of teeth of the two gears on the idler shaft was 90 and 28, respectively. The gear modulus was 1.5, and the gearbox ratio was 8.92.

The experimental data were collected at a gearbox with 0 loads and a fixed speed of 1500 rpm. The sensor used to collect the data was a piezoelectric triaxial acceleration sensor. The sensor’s sensitivity was 95.8 mV/g, and the output bias was 10.9 VDC. The gearbox failure parts used in the experiment are shown in Figure 10, containing seven single failure types, such as bearing rolling body failure, bearing outer ring failure, bearing inner ring failure, broken teeth, missing teeth, tooth surface wear, and tooth root cracking. When the data were collected, the faulty gear and bearing were installed in the second parallel shaft gearbox. The data contained 12 gearbox states, including normal conditions, seven single-fault, and four compound-fault states. The status information of the compound fault is shown in Table 1.

The gearbox vibration signal was collected by an acceleration sensor placed on the secondary parallel shaft gearbox, with a sampling frequency of 20,480 Hz and a sampling time of 8 s. For each gearbox state, the data sample contained 163,840 points. Two thousand and forty-eight points were selected as the sampling window, with five hundred overlapping points between adjacent samples. Nine hundred samples were obtained for each state sampling, and their time domain plots are shown in Figure 11. There were 10,800 samples of 12 gearbox states, and the sample details and labels are shown in Table 2. The sample dataset was randomly divided into the training, validation, and test sets according to the ratio of 8:1:1, where the training set contained 8640 samples, the validation set included 1080 samples, and the test set had 1080 samples. The training set was used to train the network and update the network parameters. The validation set was used to evaluate the model after each training round to check whether the model was overfitted on the training set. The test set was then used to assess the fault classification performance of the final model.

#### 4.1.2. Experimental Results and Analysis

To verify the effectiveness of the improved attention module CSAM proposed in this paper, the CSAM-MSCNN was compared with the CBAM-MSCNN, convolutional neural network (CNN) [30], fully-connected neural network (FNN) [30], support vector machine (SVM) [30], random forest (RF) [30], and wavelet transform and multilabel convolutional neural network (WT-MLCNN) [30]. To ensure the reliability of the comparison experiments, the network parameters of the MSCNN, the primary network structure of the CSAM-MSCNN and CBAM-MSCNN, are shown in Table 3, and the position of the attention module in the network is shown in Figure 6. The batch size was 32, the maximum number of training rounds was 100, the gradient descent optimization algorithm was chosen as Adam, the learning rate was 0.001, and the learning rate decay coefficient was 0.0001. Ten parallel experiments were performed considering the effect of random initialization parameters of the deep learning network.

Table 4 records the average classification accuracy and standard deviation of each failure mode in the test sample of the method in this paper. This paper showed excellent performance in gearbox compound faults, and the classification accuracy of all fault types was higher than 99%. The average accuracy of the overall test set was 99.61%, with a standard deviation of only 0.13%, indicating that the CSAM-MSCNN has good stability for compound fault diagnosis. Comparing the performance of the CSAM-MSCNN with the rest of the models in Table 5, it is easy to see that the classification accuracy of the proposed method in this paper is more competitive.

#### 4.1.3. Supplementary Experiments

To supplement the effectiveness of the proposed method, simulation data were used to test the diagnostic performance of the proposed model. The structure of the gearbox simulation model is shown in Figure 9. The simulated gearbox was set to operate at 0 loads and 1500 rpm, and the information on the dataset used for the experiment is shown in Table 6. The data had four gearbox states, including the normal state, two single-fault states, and the compound fault state coupled by a single fault, and the vibration signal time domain diagram is shown in Figure 12. The original vibration data were sampled by sliding windows with 2048 and 500 window shift lengths. Five hundred samples were sampled for each state to form the sample dataset. The dataset was divided into the training set, validation set, and test set according to the ratio of 8:1:1, i.e., the training set contained 1600 samples, and the validation set and test set had 200 samples each.

The output layer dimension of the network was changed to three, and the rest of the model parameters are shown in Table 3. The maximum number of training rounds was 20. The rest of the hyperparameter settings were the same as in Experiment 1. The training process of the CSAM-MSCNN is shown in Figure 13, and the network converged after five training rounds. Since depth model parameters are affected by random initialization, ten parallel experiments were conducted to calculate the average accuracy and standard deviation. The results are shown in Table 7. Through 10 parallel experiments, the average accuracy of the CSAM-MSCNN on the test set was 99.46%, with a standard deviation of 0.67%. To visualize the fault classification effect on the test set, t-distributed stochastic neighbor embedding (t-SNE) was used to map the output layer features of the network to a two-dimensional plane, as shown in Figure 14. It can be seen that the CSAM-MSCNN can easily distinguish the four gearbox operating states. The results of this experiment show that the CSAM-MSCNN can perform the diagnosis task well for both the actual working condition data and the simulation data, which further verifies the effectiveness of the proposed method.

### 4.2. Experiment 2: 2009 PHM Gearbox Fault Diagnosis

#### 4.2.1. Description of Experimental Data

The data for this experiment originated from the 2009 PHM Challenge, which are representative of general industrial gearbox data. The gearbox contained three shafts: the input shaft; the idler shaft; and the output shaft. It also included six bearings and four gears, divided into spur and helical gears. In this paper, the data of the spiral gear set was selected, and only the first channel of the vibration signal was used. A sketch of the gearbox structure is shown in Figure 15, with 16 and 40 teeth on the input and output shafts, respectively, and 48 and 24 teeth on the two gears of the idler shaft.

The experimental dataset was collected from the gearbox at low load and 30, 40, and 50 Hz speed and contained six different health states. The six states of the gearbox are shown in Table 8, having eight independent states of the normal, broken tooth, missing tooth, bearing compound fault, bearing inner ring fault, bearing outer ring fault, shaft bending, and shaft unbalance, so the number of neurons in the final output layer of the CSAM-MSCNN was 8. The sampling frequency was 66.7 kHz, the sampling time was 4 s, the selected sample window length was 2048, and the window moving step was 500. Six hundred samples were collected for each health state at 30, 40, and 50 Hz speed as the network dataset, so the total dataset contained 10,800 samples. The dataset was randomly divided into the training set, validation set, and test set in the ratio of 8:1:1, and the samples and labels are described in detail, as shown in Table 9.

#### 4.2.2. Experimental Results and Analysis

To further show the superiority of the method in this paper for composite fault diagnosis, it was compared with the CBAM-MSCNN, CNN, FNN, SVM, RF, and WT-MLCNN. The detailed network parameters of the CSAM-MSCNN are shown in Table 3. Similarly, the network parameters of the CBAM-MSCNN remained the same as in the CSAM-MSCNN, except for the attention module. The batch size was 32, the maximum number of training rounds was 200, the gradient descent optimization algorithm was chosen as Adam, the learning rate was 0.001, and the learning rate decay coefficient was 0.0001. The training process of the CSAM-MSCNN is shown in Figure 16.

Ten parallel experiments were conducted to avoid specificity and chance, and the experimental results are shown in Table 10 and Table 11. The results when comparing the method in this paper with the CBAM-MSCNN and the rest of the models are shown in Figure 17. 

To demonstrate the advantages of the proposed CSAM-MSCNN more intuitively, feature visualization was implemented using t-SNE. The features extracted at the first scale, the features extracted at the second scale, the features extracted at the third scale, the features obtained by feature fusion, the features corrected by the attention module, and the last fully connected layer features of the output layer were mapped to two-dimensional space, in the model. The output features of each network layer are shown in Figure 18.

Analysis of the above experimental results led to the following:

(1) In Figure 16, after 200 training iterations, the loss function and accuracy curves of the training and validation set finally converged. This indicates that CSAM-MSCNN can thoroughly explore the composite fault-related information in the gearbox vibration signal and achieve accurate fault classification.

(2) Comparing the results of the 10 parallel experiments of the CSAM-MSCNN and CBAM-MSCNN in Table 10 and Table 11. The average test accuracy of the CSAM-MSCNN was 99.61%, with a standard deviation of 0.13. In comparison, the average test accuracy of the CBAM-MSCNN was 99.22% with a standard deviation of 0.37%, which indicates that the improvement in the CSAM is effective. The main reason is that the CSAM uses a shared one-dimensional convolution instead of the shared fully connected layer in CBAM. This reduces the model parameters, solves the side effects generated by the original bottleneck structure, and improves the model’s robustness.

(3) The advantages of the proposed method in terms of diagnostic accuracy and stability can be visualized in Figure 17. Notably, traditional models such as CNN, FNN, SVM, and RF use single-label classifiers and perform poorly when diagnosing raw vibration data. The proposed method used a multilabel classifier to diagnose end-to-end faults using primary vibration data, which can decouple compound faults into multiple single fault modes while obtaining high diagnostic accuracy. The fault diagnosis task in this paper was more complex than traditional fault diagnosis tasks because we were diagnosing multiple faults simultaneously rather than treating them as a new failure mode. In contrast, compared with WT-MLCNN and CBAM-MSCNN, which also use multilabel classification, the CSAM-MSCNN showed more competitive diagnostic accuracy and standard deviation.

(4) As seen from the plots (a), (b), and (c) in Figure 18, the clustering effect of the proposed features in the CSAM-MSCNN varied among the three scales, which indicates that the use of multiscale network extraction results can enrich the fault information learned by the network. Observing plot (d), it can be found that performing multiscale feature fusion can yield better clustering results than single-scale feature extraction. This validates that fusing features from the three scales is effective and can complement each other’s feature information that needs to be focused on each scale. By comparing (d) and (c) plots, we found that the feature maps corrected by the CSAM were better clustered than those obtained by feature fusion. This indicates that the CSAM can effectively deal with the feature redundancy problem brought by multiscale feature fusion, focus the network attention on critical fault information, highlight the fault features that play a crucial role in diagnostic decisions, and suppress the features that are meaningless for fault classification or even lead to classification errors. The final (f) plot shows that the model can achieve good fault signal classification and that the CSAM-MSCNN can achieve end-to-end fault diagnosis using raw time-domain vibration signals.

## 5. Conclusions

This paper proposed a new gearbox composite fault diagnosis method, CSAM-MSCNN. The technique combined an improved channel–space attention module, a multiscale network structure, and a multilabel classifier, which can take full advantage of the feature extraction capability of the multiscale system, the feature enhancement capability of the attention mechanism, and the edges of the multilabel classifier in handling compound faults. The proposed method’s validity was verified using gearbox data collected in the laboratory and the data from the 2009 PHM Challenge. In Experiment 1, the average accuracy of the CSAM-MSCNN was 99.79%, with a standard deviation of only 0.13%. In Experiment 2, the average accuracy of the CSAM-MSCNN was 99.61%, with a standard deviation of only 0.13%. In two experiments, the proposed method in this paper effectively diagnosed compound faults in gearboxes and showed a more competitive fault diagnosis performance than other models. The method proposed in this paper is mainly used for severe gearbox failures. The intention is to apply the method to early faint fault diagnosis in future research.

## Figures and Tables

**Figure 1 sensors-23-03827-f001:**
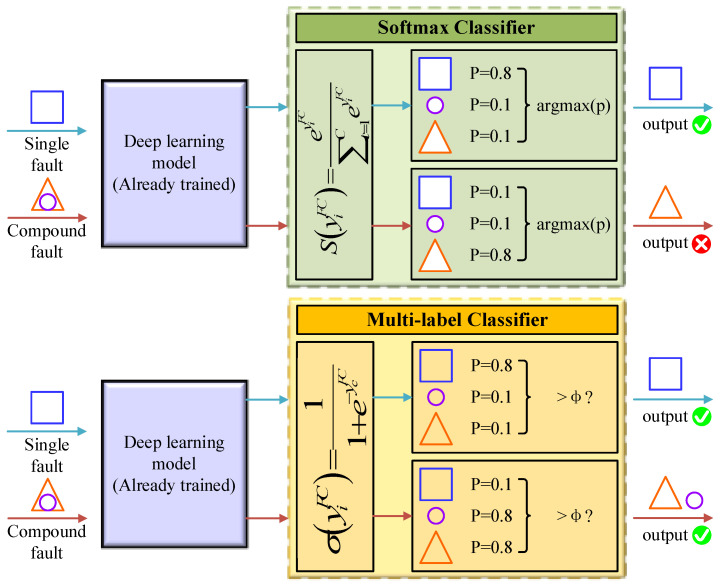
SoftMax classifier and multilabel classifier.

**Figure 2 sensors-23-03827-f002:**
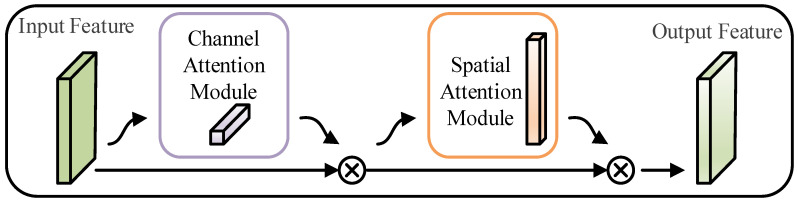
Structure of the CBAM.

**Figure 3 sensors-23-03827-f003:**
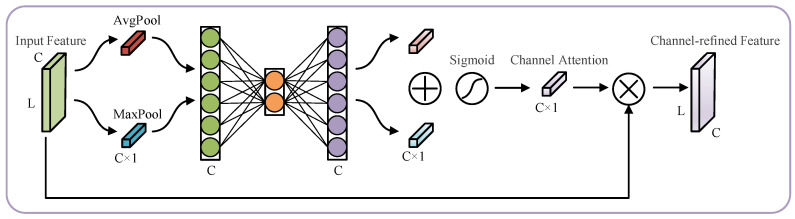
Channel Attention Module in the CBAM. C stands for channel and L stands for length.

**Figure 4 sensors-23-03827-f004:**
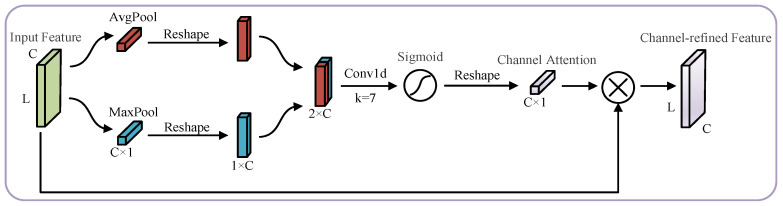
Improved Channel Attention Module.

**Figure 5 sensors-23-03827-f005:**
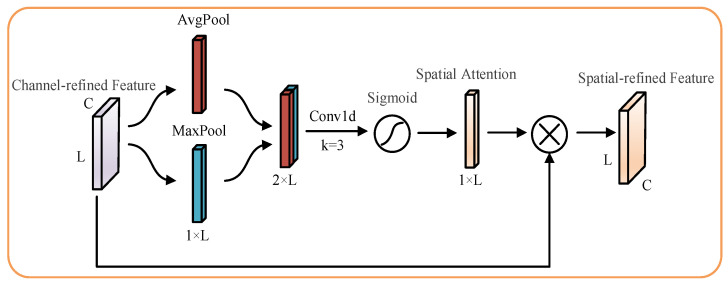
Spatial Attention Module in the CSAM.

**Figure 6 sensors-23-03827-f006:**
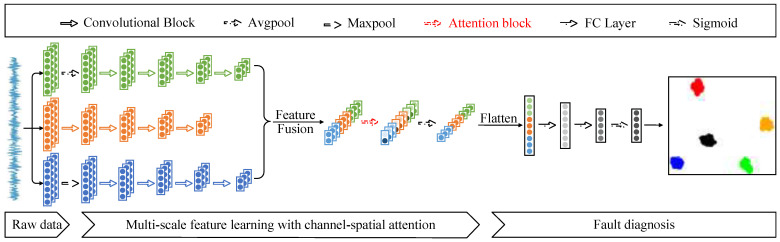
CSAM-MSCNN architecture diagram.

**Figure 7 sensors-23-03827-f007:**
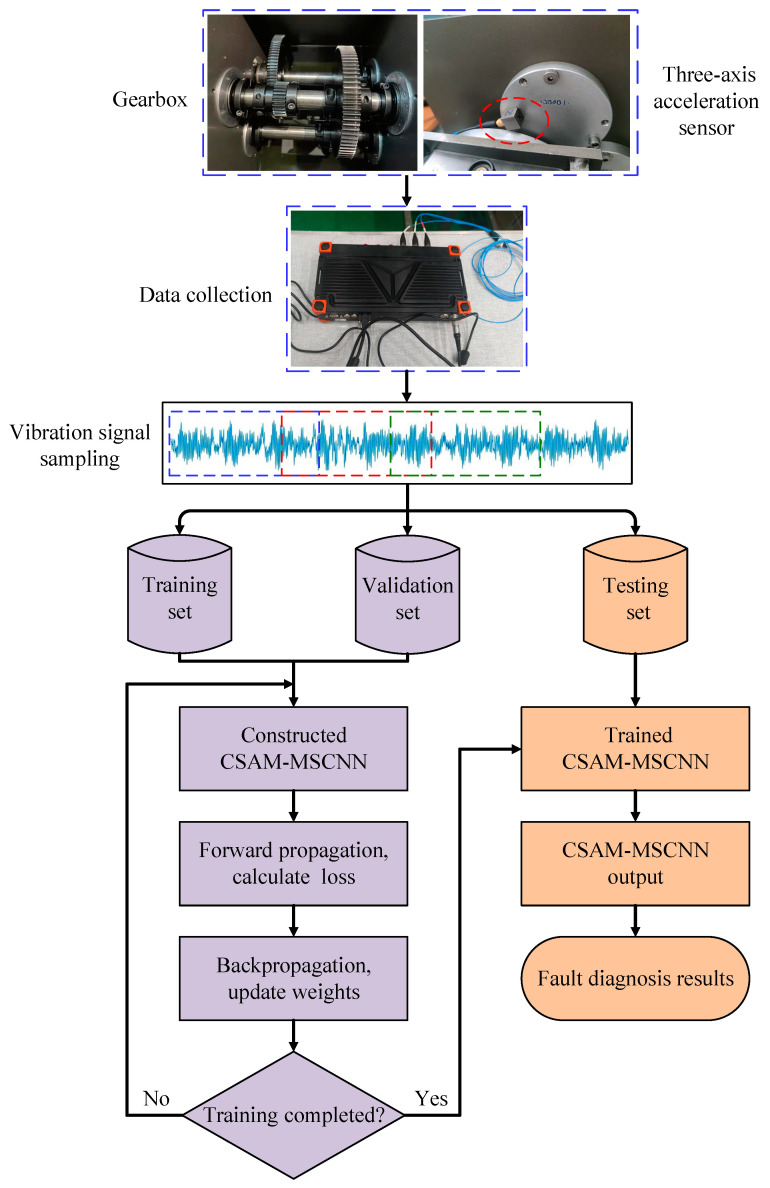
Fault diagnosis process based on the CSAM-MSCNN.

**Figure 8 sensors-23-03827-f008:**
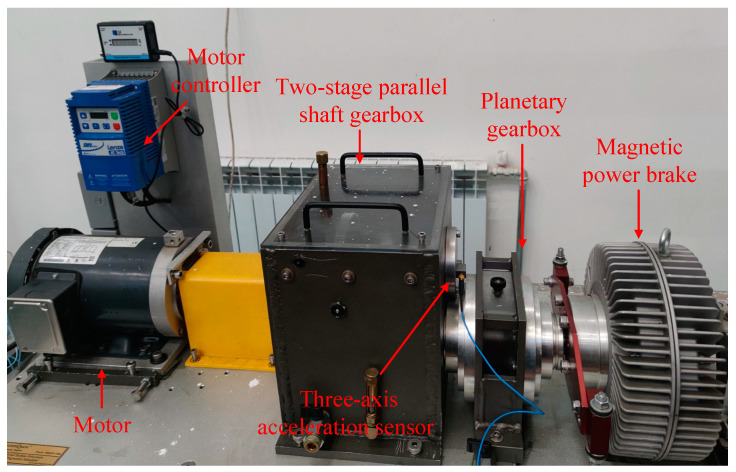
Wind turbine drive system fault diagnosis test bench.

**Figure 9 sensors-23-03827-f009:**
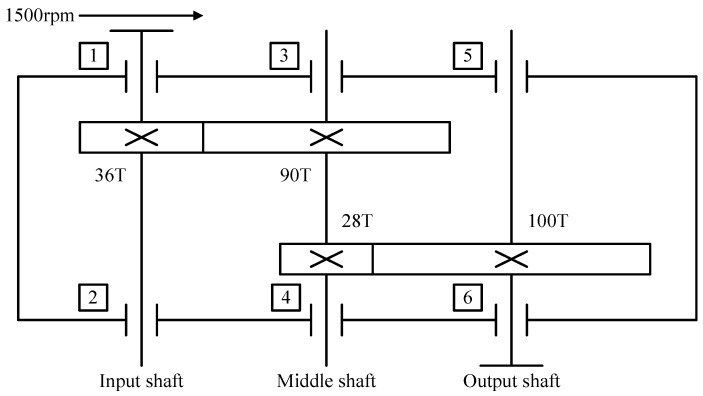
Structure diagram of parallel shaft gearbox.

**Figure 10 sensors-23-03827-f010:**
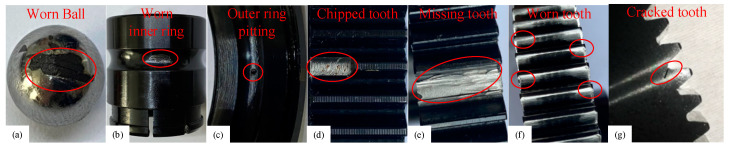
Faulty parts in the gearbox. (**a**) Bearing rolling body wear failure; (**b**) Bearing inner ring wear failure; (**c**) Bearing outer ring pitting failure; (**d**) Gear tooth breakage failure; (**e**) Missing gear teeth failure; (**f**) Gear tooth wear failure; (**g**) Gear tooth root crack failure.

**Figure 11 sensors-23-03827-f011:**
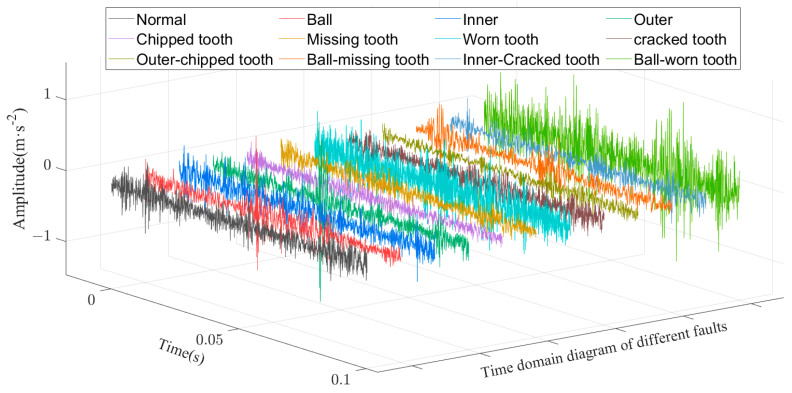
Time domain diagram of the vibration signal.

**Figure 12 sensors-23-03827-f012:**
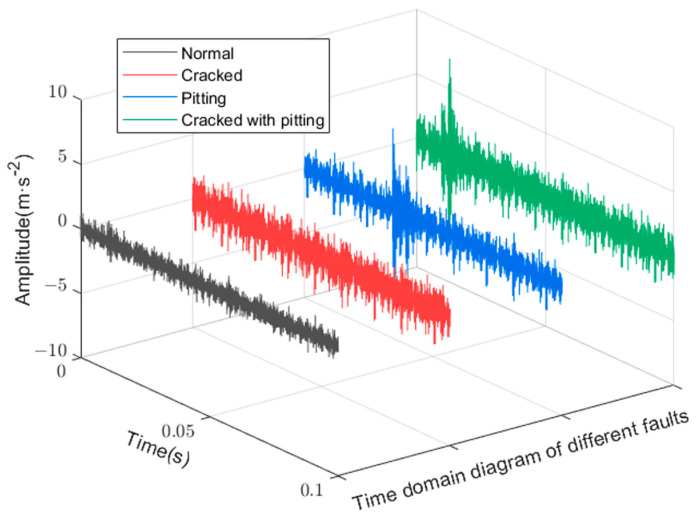
Time domain diagram of the simulated signal.

**Figure 13 sensors-23-03827-f013:**
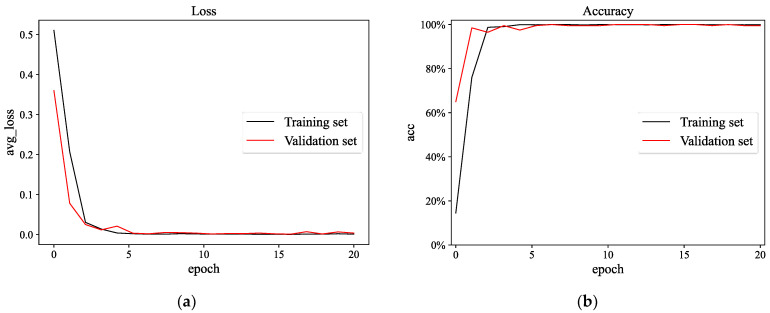
Simulation experimental training process of CSAM-MSCNN: (**a**) Loss function curve; (**b**) accuracy curve.

**Figure 14 sensors-23-03827-f014:**
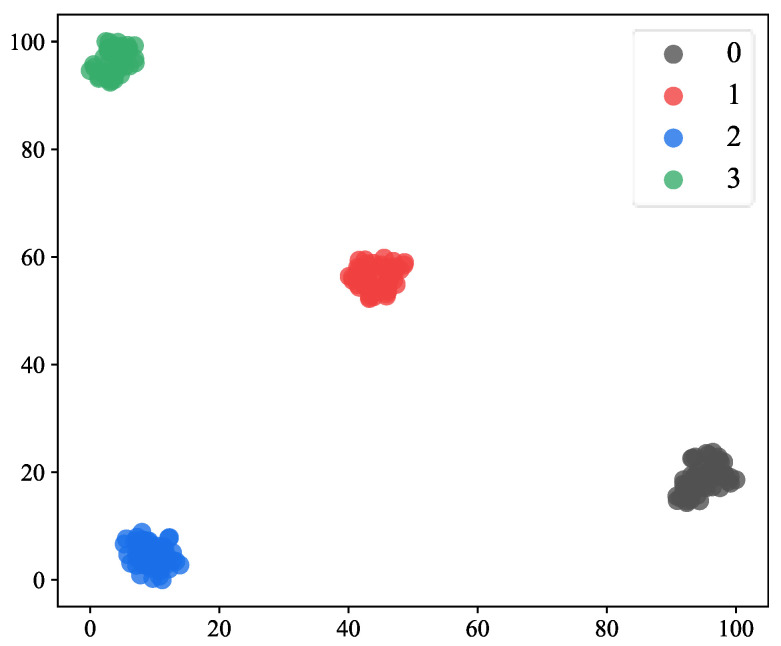
Visualization of test set diagnostic results. Labels 0, 1, 2, 3 represent normal, cracked tooth, pitting tooth, and gear crack-pitting compound failure, respectively.

**Figure 15 sensors-23-03827-f015:**
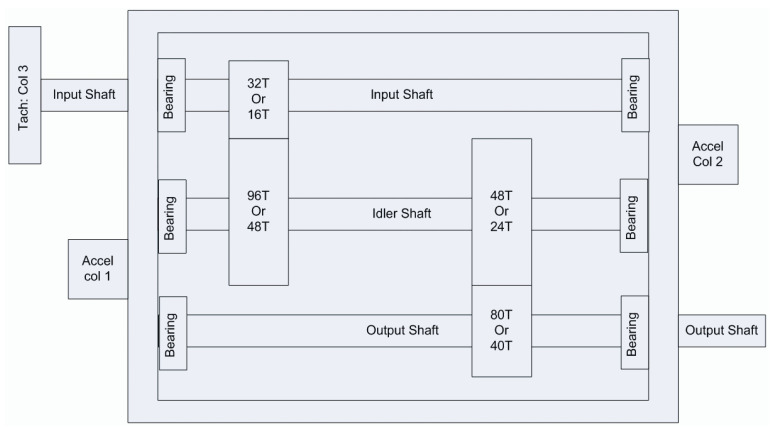
Schematic of the gearbox used in the PHM 2009 Challenge Data [32].

**Figure 16 sensors-23-03827-f016:**
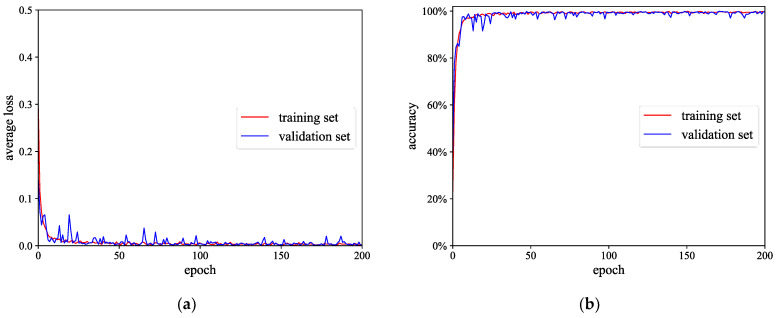
The training process of the CSAM-MSCNN: (**a**) Loss function curve; (**b**) accuracy curve.

**Figure 17 sensors-23-03827-f017:**
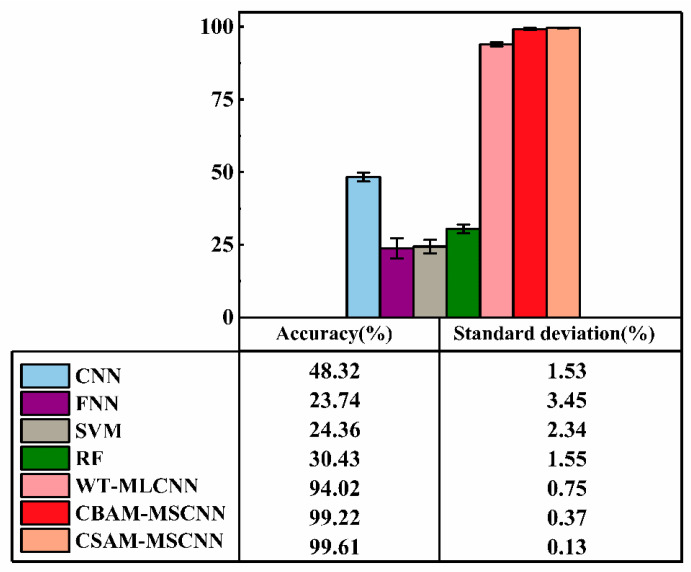
Comparison of accuracy and standard deviation between the CSAM-MSCNN and other models.

**Figure 18 sensors-23-03827-f018:**
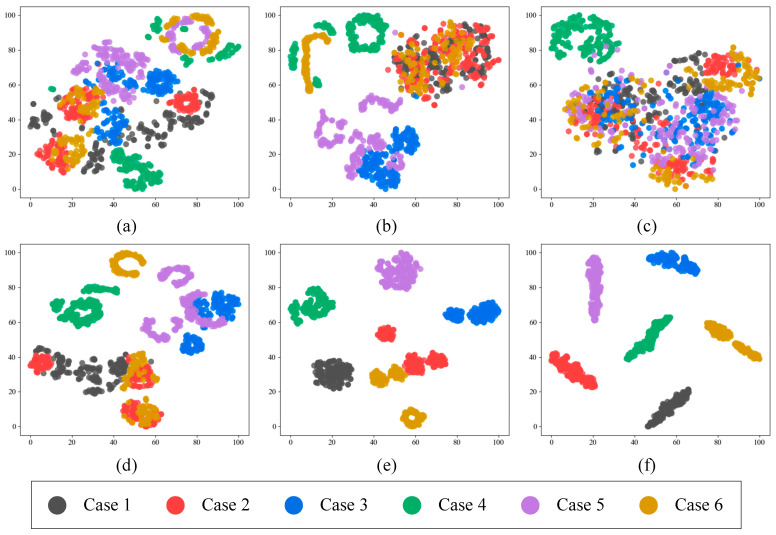
t-SNE visualization of network layer features: (**a**) Features extracted at the first scale; (**b**) Features extracted at the second scale; (**c**) Features extracted at the third scale; (**d**) Features obtained from feature fusion; (**e**) Features after CSAM correction; (**f**) The last layer features of the output layer.

**Table 1 sensors-23-03827-t001:** The information on the compound fault status.

Compound Fault (A-B)	Outer-Chipped Tooth	Ball-Missing Tooth	Inner-Cracked Tooth	Ball-Worn Tooth
Single fault types	Outer ring pitting, chipped tooth	Worn ball, missing tooth	Worn inner ring, cracked tooth	Worn ball, worn tooth
Fault information	Outer ring pitting: moderate pitting, with a pitting diameter of 1 mm, located in the center of the raceway. Chipped tooth: A gear tooth is missing a quarter, i.e., half in each direction of tooth height and tooth width.	Worn ball: moderate wear, the wear area is about 3 mm long by 1 mm wide irregular shape. Missing tooth: a wheel tooth is completely broken, i.e., one tooth is missing.	Worn inner ring: Moderate wear, the wear area is an irregular shape about 3 mm long and 1 mm wide, located in the center of the raceway. cracked tooth: Crack depth 1.5 mm	Worn ball: moderate wear, the wear area is about 3 mm long by 1 mm wide irregular shape. worn tooth: Moderate grinding of all gear teeth
Fault location	Faulty bearing in position 6; The faulty gear is located in the first-stage driven wheel The faulty gear is a first-class driven wheel.	Faulty bearing in position 6; The faulty gear is located in the primary active wheel.	Faulty bearing in position 6; The faulty gear is located in the secondary driven wheel.	Faulty bearing in position 6; The faulty gear is located in the secondary active wheel.

**Table 2 sensors-23-03827-t002:** Experimental samples and labels.

Work Condition	Length of Sample	Number of Samples	Label Vector
Normal	2048	900	[1, 0, 0, 0, 0, 0, 0, 0]
Ball	2048	900	[0, 1, 0, 0, 0, 0, 0, 0]
Inner	2048	900	[0, 0, 1, 0, 0, 0, 0, 0]
Outer	2048	900	[0, 0, 0, 1, 0, 0, 0, 0]
Chipped tooth	2048	900	[0, 0, 0, 0, 1, 0, 0, 0]
Missing tooth	2048	900	[0, 0, 0, 0, 0, 1, 0, 0]
Worn tooth	2048	900	[0, 0, 0, 0, 0, 0, 1, 0]
Cracked tooth	2048	900	[0, 0, 0, 0, 0, 0, 0, 1]
Outer with chipped tooth	2048	900	[0, 0, 0, 1, 1, 0, 0, 0]
Ball with missing tooth	2048	900	[0, 1, 0, 0, 0, 1, 0, 0]
Inner with cracked tooth	2048	900	[0, 0, 1, 0, 0, 0, 0, 1]
Ball with worn tooth	2048	900	[0, 1, 0, 0, 0, 0, 1, 0]

**Table 3 sensors-23-03827-t003:** Detailed network parameters of the CSAM-MSCNN.

Block	Layer Name	Kernel Size/Stride/Channel	Output Shape (C, L)
\	Input layer	\	1 × 2048
\	Adaptive average pool	\	1 × 512
Convolutional block1-1	Conv1d	16/4/16	16 × 128
	Max pool	2/2/16	16 × 64
Convolutional block1-2	Conv1d	3/1/32	32 × 64
	Max pool	2/2/32	32 × 32
Convolutional block1-3	Conv1d	3/1/64	64 × 32
	Max pool	2/2/64	64 × 16
Convolutional block1-4	Conv1d	3/1/64	64 × 16
	Max pool	2/2/64	64 × 8
Convolutional block2-1	Conv1d	64/16/16	16 × 128
	Max pool	2/2/16	16 × 64
Convolutional block2-2	Conv1d	3/1/32	32 × 64
	Max pool	2/2/32	32 × 32
Convolutional block2-3	Conv1d	3/1/64	64 × 32
	Max pool	2/2/64	64 × 16
Convolutional block2-4	Conv1d	3/1/64	64 × 16
	Max pool	2/2/64	64 × 8
\	Adaptive max pool	\	1 × 512
Convolutional block3-1	Conv1d	16/4/16	16 × 128
	Max pool	2/2/16	16 × 64
Convolutional block3-2	Conv1d	3/1/32	32 × 64
	Max pool	2/2/32	32 × 32
Convolutional block3-3	Conv1d	3/1/64	64 × 32
	Max pool	2/2/64	64 × 16
Convolutional block3-4	Conv1d	3/1/64	64 × 16
	Max pool	2/2/64	64 × 8
\	Feature fusion	\	192 × 8
CSAM	\	\	192 × 8
\	Global average pool	\	192 × 1
Classification block	Flatten	\	192
	Linear	\	64
	Linear	\	8

**Table 4 sensors-23-03827-t004:** Results of ten parallel experiments of CSAM-MSCNN.

Gearbox Status	1	2	3	4	5	6	7	8	9	10	Accuracy ± Std (%)
Normal	100	96.51	100	100	100	100	100	100	100	98.78	99.52 ± 1.07
Ball	100	100	100	100	100	100	98.84	100	100	100	99.88 ± 0.34
Inner	100	100	100	98.90	100	97.7	100	98.96	100	100	99.55 ± 0.74
Outer	100	100	100	100	97.62	98.8	100	98.97	98.89	100	99.42 ± 0.78
Chipped tooth	100	100	100	100	100	100	100	100	100	100	100 ± 0
Missing tooth	100	100	100	100	100	100	100	100	100	100	100 ± 0
Worn tooth	100	100	100	100	100	100	100	100	100	100	100 ± 0
Cracked tooth	100	100	100	100	100	100	100	100	100	100	100 ± 0
Outer with chipped tooth	100	100	100	100	100	100	100	100	100	100	100 ± 0
Ball with missing tooth	98.90	100	97.96	100	95.06	100	100	100	100	100	99.19 ± 1.52
Inner with cracked tooth	100	100	100	100	100	100	100	100	100	100	100 ± 0
Ball with worn tooth	100	100	100	100	100	100	100	100	100	100	100 ± 0
Testing set	99.90	99.72	99.81	99.9	99.44	99.72	99.90	99.81	99.90	99.81	99.79 ± 0.13

**Table 5 sensors-23-03827-t005:** Mean test accuracy and standard deviation of the proposed method versus other methods.

Methods	Classification	Accuracy ± Std (%)
CNN	SC	76.50 ± 1.51
FNN	SC	21.58 ± 4.81
SVM	SC	24.29 ± 2.73
RF	SC	44.62 ± 1.83
WT-MLCNN	MLC	95.48 ± 0.81
CBAM-MSCNN	MLC	98.62 ± 0.38
CSAM-MSCNN	MLC	99.79 ± 0.13

**Table 6 sensors-23-03827-t006:** Description of gearbox failure simulation dataset.

Work Condition	Fault Information	Length of Sample	Number of Samples	Label Vector
Normal	\	2048	500	[1, 0, 0]
Cracked tooth	The gear is a first-class passive wheel, and the crack depth is 2 mm.	2048	500	[0, 1, 0]
Pitting tooth	The gear is a first-class active wheel with severe pitting.	2048	500	[0, 0, 1]
Cracked tooth- pitting tooth	Compound failure of tooth root crack and tooth surface pitting	2048	500	[0, 1, 1]

**Table 7 sensors-23-03827-t007:** Parallel experimental results of simulation data for CSAM-MSCNN.

Gearbox Status	1	2	3	4	5	6	7	8	9	10	Accuracy ± Std (%)
Normal	100	100	100	100	100	100	100	100	100	100	100 ± 0
Cracked tooth	100	100	96.61	100	100	98	100	100	100	100	99.88 ± 1.12
pitting tooth	100	100	100	100	100	100	100	100	100	100	100 ± 0
Cracked tooth with pitting tooth	92.86	100	100	100	93.10	100	100	98	100	100	98.39 ± 2.77
Testing set	98.21	100	99.15	100	98.27	99.5	100	99.5	100	100	99.46 ± 0.67

**Table 8 sensors-23-03827-t008:** Gearbox Health Status Description.

Case	Gear				Bearing						Shaft	
	16T	48T	24T	40T	IS:IS	ID:IS	OS:IS	IS:OS	ID:OS	OS:OS	Input	Output
Case1	Good	Good	Good	Good	Good	Good	Good	Good	Good	Good	Good	Good
Case2	Good	Good	**Chipped**	Good	Good	Good	Good	Good	Good	Good	Good	Good
Case3	Good	Good	**Broken**	Good	Good	Good	Good	**Combination**	**Inner**	Good	**Bent Shaft**	Good
Case4	Good	Good	Good	Good	Good	Good	Good	**Combination**	**Ball**	Good	**Imbalance**	Good
Case5	Good	Good	**Broken**	Good	Good	Good	Good	Good	**Inner**	Good	Good	Good
Case6	Good	Good	Good	Good	Good	Good	Good	Good	Good	Good	**Bent Shaft**	Good

**Table 9 sensors-23-03827-t009:** Sample and Label Instructions.

Work Condition	Length of Sample	Number of Samples	Label Vector
Case 1	2048	1800	[1, 0, 0, 0, 0, 0, 0, 0]
Case 2	2048	1800	[0, 1, 0, 0, 0, 0, 0, 0]
Case 3	2048	1800	[0, 0, 1, 1, 1, 0, 1, 0]
Case 4	2048	1800	[0, 0, 0, 1, 0, 1, 0, 1]
Case 5	2048	1800	[0, 0, 1, 0, 1, 0, 0, 0]
Case 6	2048	1800	[0, 0, 0, 0, 0, 0, 1, 0]

**Table 10 sensors-23-03827-t010:** Parallel experimental results of the CSAM-MSCNN.

Gearbox Status	1	2	3	4	5	6	7	8	9	10	Accuracy ± Std (%)
Case 1	100	100	100	100	100	100	100	100	99.40	100	99.94 ± 0.18
Case 2	100	98.94	99.47	100	99.47	100	99.47	97.35	99.47	99.47	99.36 ± 0.74
Case 3	100	99.41	100	100	100	100	100	100	99.41	100	99.88 ± 0.23
Case 4	100	100	100	100	100	100	100	100	100	100	100 ± 0
Case 5	100	100	100	99.02	99.51	99.51	99.51	99.51	100	99.02	99.61 ± 0.36
Case 6	98.84	98.84	97.11	99.42	98.84	98.84	99.84	99.42	100	98.84	98.99 ± 0.75
Testing set	99.81	99.54	99.44	99.72	99.63	99.72	99.63	99.35	99.72	99.54	99.61 ± 0.13

**Table 11 sensors-23-03827-t011:** Parallel experimental results of the CBAM-MSCNN.

Gearbox Status	1	2	3	4	5	6	7	8	9	10	Accuracy ± Std (%)
Case 1	100	100	100	98.81	100	100	98.81	97.02	99.4	100	99.40 ± 0.92
Case 2	97.35	98.94	97.88	100	100	98.41	96.3	98.41	99.47	94.71	98.14 ± 1.59
Case 3	100	100	100	100	100	100	100	100	99.41	100	99.94 ± 0.17
Case 4	100	100	100	100	100	100	100	99.44	100	100	99.94 ± 0.16
Case 5	100	100	99.51	100	99.51	100	100	98.53	100	100	99.75 ± 0.16
Case 6	93.06	98.27	98.27	97.11	98.27	98.84	99.42	100	99.42	98.84	98.15 ± 1.86
Testing set	98.43	99.54	99.26	99.35	99.63	99.54	99.07	98.89	99.63	98.89	99.22 ± 0.37

## Data Availability

Not applicable.

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
