# Peer review of "Multiscale Convolutional Neural Network Based on Channel Space Attention for Gearbox Compound Fault Diagnosis"

_sensors, 2023, doi:10.3390/s23083827_

Round 1

Reviewer 1 Report

In this paper, a gearbox compound fault diagnosis method is proposed, and the channel-space attention is used to solve the problem that the information interaction between channels is ignored.  The fault diagnosis method is proposed based on multi-scale convolution neural network and channel-space attention. This method is more reliable for the diagnosis of gearbox and has certain academic significance, however, there are the following problems: 1) What is the connection between Fig.3, Fig.4, Fig.5 and Fig.6? 2) The validation set does not participate in parameter update in page 10, however, the training set and the validation set are used to train the model, is this correct? 3) The authors mentioned the data contain four compound-fault states in page 11, please explain the information of the four compound-fault states. 4) The vibration signal collected by the sensor contains a lot of noise, but no noise reduction is carried out in the case study. Is the influence of the noise signal on the accuracy of fault diagnosis taken into account?

Reviewer 2 Report

A gearbox compound fault diagnosis method is proposed to address the multiple single faults problems in this paper. Multi-scale convolutional neural network is used to effectively mine the compound fault in-formation from vibration signals. A channel-space attention module is proposed which is embedded into the MSCNN to assign weights to multi-scale features for enhancing the feature differentiation processing ability of MSCNN. A multi-label classifier is used to outputs single or multiple labels for recognizing single or compound faults. The effectiveness of the method was verified with two gearbox data-sets. Some interesting conclusions have been obtained.

1. The advantages of proposed method should clearly be described and the comparisons with traditional method should also be provided.

2. Figure 1 should be clearly provided.

3. Simulation analysis and results on the new method can be appropriately supplemented.

Reviewer 3 Report

Could you describe the characteristics of the vibration sensor, why was this type of sensor chosen?
Why three sensors are used, and how does it affect the location of the sensors, it is not clear how the signals from the three sensors are mixed, and if only one sensor is used.
What is the signal conditioning stage?
It is unclear how the signals from the three sensors are mixed and if only one is used.
Does the methodology work using another type of signal, e.g. current?
How does it affect the sampling frequency, why is 20480 Hz used in one case and 66.7Hz in another?
Is not the frequency too low in the case of 66.7Hz?
Mechanically where should the fault frequencies be located?
In the conclusions section, the authors mention that the proposed methodology only works for severe cases, and the state of the art is incipient damage. How do they justify their contribution?

Round 2

Reviewer 3 Report

The answer to question 1 is not the expected one. I don't want you to define what an accelerometer is; I want you to describe quantitatively the characteristics of the sensor used.
Same case for question 3, do not define the conditioning stage; describe the one you are using.
Question 6, in the literature, you can find the frequencies in which the failure occurs. For your case, mathematically, in which frequency should the damage be seen?
In the case of severe and incipient damage, I agree that there is no conflict between them. However, if I do the detection in the initial stage, why would I need the severe one if I have already detected it previously?

Round 3

Reviewer 3 Report

Thank you for the answers